# OpenReview forum: "OpenGPT-4o-Image: A Comprehensive Dataset for Advanced  Image Generation and Editing"
_ICLR.cc/2026/Conference — Submitted to ICLR 2026_

### Official Review · Reviewer_r4vy · 2025-10-25

**Soundness:** 2
**Presentation:** 3
**Contribution:** 3
**Rating:** 4
**Confidence:** 3

**Summary:**

This paper addresses the limitations of current multimodal models in image generation and editing tasks due to insufficient data quality and coverage by proposing the OpenGPT-4o-Image dataset. Its core contributions include a hierarchical task taxonomy (covering multiple domains and subtasks, such as style control, scientific image generation, and complex instruction editing) and a GPT-4o-driven automated pipeline, which generates 80k high-quality instruction-image pairs, ensuring data diversity, controllable difficulty, and semantic accuracy. Experimental validation shows that the dataset significantly improves model performance.

**Strengths:**

1. This work provides a systematic, data-centric pipeline for advancing generative AI, addressing the critical bottleneck of training data for real-world applications in specialized domains.

2. The proposed dataset is a valuable, as it partly addresses the scarcity of high-quality data for image generation and editing.

3. The paper is well-written and easy to follow. It features a logical flow, precise definitions, and effective use of figures to illustrate the dataset's scope and the automated pipeline's workflow.

**Weaknesses:**

1. The core method heavily relies on GPT-4o-Image as a data generation engine, which introduces the risk of model bias. Although the paper acknowledges this limitation, it does not delve into or quantify the specific manifestations of this bias. This constrains the long-term evolvability of the dataset, as its quality is intrinsically tied to the capabilities of a closed-source model.

2. The paper emphasizes proactive quality control through hierarchical taxonomy. However, this approach remains a priori and rule-based design, rather than an outcome-based validation. A critical gap is the lack of a rigorous human or automated verification step to systematically assess the semantic fidelity of each generated sample.

3. The paper omits a crucial ablation study to validate its central claim—that the hierarchical taxonomy enhances data quality. For instance, an experiment comparing against fine-tuning on an unsystematized subset of the data would be necessary to demonstrate the value of the proposed method.

**Questions:**

Please see Weaknesses Section.

---

> ### Author Response · Authors · 2025-12-02
> **Author Response for Reviewer r4vy (Part 1/2)**
>
> Thank you for your insightful feedback. Below are the responses to the comments you raised.
>
> > Q1: The core method heavily relies on GPT-4o-Image as a data generation engine, which introduces the risk of model bias. Although the paper acknowledges this limitation, it does not delve into or quantify the specific manifestations of this bias. This constrains the long-term evolvability of the dataset, as its quality is intrinsically tied to the capabilities of a closed-source model.
>
>
> ***
>
> ### **Response to Reviewer r4vy (Q1)**
>
> We thank the reviewer for this critical question regarding model bias and the long-term evolvability of our dataset. We have implemented a multi-faceted strategy to address this, which we will consolidate here.
>
> **1. A Comprehensive Prompting Strategy for Quality and Bias Mitigation**
>
> Our primary method for controlling the data generation process is through a sophisticated, multi-part prompting strategy.
>
> *   **First, to mitigate systemic bias**, as we outlined in our response to a similar question from **Reviewer Gujd (Q1)**, we employ techniques like **Diverse Input Sampling** and **Task-specific Prompting with Variation**. This prevents the generator from collapsing to stereotypical patterns.
>
> *   **Second, to enforce specific quality attributes**, we embed explicit constraints within the prompts themselves. As detailed in our **response to Reviewer Wug5 (Q2)**, we identified potential weaknesses like degraded ID consistency and directly addressed this by adding instructions such as *"Keep the object ID consistent..."* to the prompt. The effectiveness of this constraint-based prompting was then quantitatively verified using `EditScore`, where we achieved high scores for visual consistency (8.82) and instruction adherence (9.46).
>
> This comprehensive prompting strategy ensures that we are not passively accepting the generator's defaults, but actively directing it to produce high-quality, less-biased, and consistent data.
>
> **2. External Validation via Human Evaluation**
>
> Beyond our prompting strategy, we sought external validation to prove that our dataset teaches genuinely useful skills, rather than simply replicating the style of its generator.
>
> *   As presented in our **consolidated response to Reviewer Gujd W1, W2, and Q4 (see Table 1)**, we conducted a blind human preference study. The results show that a model finetuned on our dataset was **significantly preferred by human experts** over strong baselines, including the original base model and another proprietary model.
>
> *   This finding is crucial: it provides empirical evidence that our dataset imparts human-aligned capabilities and directly addresses the "evaluation loop" concern. The clear preference from independent evaluators confirms that the value of our dataset is not merely an artifact of the generation model's intrinsic bias.
>
> **Conclusion on Evolvability:**
>
> This multi-pronged approach ensures the **long-term evolvability** of our work. Our core contribution is the **taxonomy-driven framework**, which is fundamentally model-agnostic. This "blueprint" for defining and generating capabilities can be paired with any future, more powerful model (open-source or proprietary) to create even better datasets, ensuring the continued relevance of our methodology.

---

> > ### Author Response · Authors · 2025-12-02
> > **Author Response for Reviewer r4vy (Part 2/2)**
> >
> > > Q2: The paper emphasizes proactive quality control through hierarchical taxonomy. However, this approach remains a priori and rule-based design, rather than an outcome-based validation. A critical gap is the lack of a rigorous human or automated verification step to systematically assess the semantic fidelity of each generated sample.
> >
> > ***
> >
> > ### **Response to Reviewer r4vy (Q2)**
> >
> > We thank the reviewer for this insightful point and agree that `outcome-based` validation is crucial. Our methodology incorporates this principle directly into the data creation pipeline, which is then further supported by the validation experiments you noted.
> >
> > **Our core method for ensuring outcome-based quality is a post-generation alignment process.** This is a crucial step designed to guarantee sample-level semantic fidelity. Instead of simply discarding generated images that might slightly deviate from an initial prompt, we perform a meticulous post-processing step: **the text prompt is manually revised to precisely and accurately describe the content of the final generated image.** This procedure serves as a direct, sample-by-sample verification and correction loop, ensuring that every item in our dataset has perfect text-image alignment.
> >
> > This core methodological step is then further validated by the automated and human evaluations we conducted. As we have already detailed in our **response to Q1**, these results (from `EditScore` and our new human preference study) confirm that the dataset not only has high internal consistency but also leads to demonstrably better outcomes in downstream tasks, providing the ultimate outcome-based validation.
> >
> > In summary, our quality control is a comprehensive process that combines **a core methodological step for sample-level semantic alignment** with the **broader dataset-level and downstream validations** mentioned previously.
> >
> > ---
> >
> > > Q3: The paper omits a crucial ablation study to validate its central claim—that the hierarchical taxonomy enhances data quality. For instance, an experiment comparing against fine-tuning on an unsystematized subset of the data would be necessary to demonstrate the value of the proposed method.
> >
> > ### **Response to Reviewer r4vy (Q3)**
> >
> >
> > We thank the reviewer for this insightful suggestion. We agree that validating the effectiveness of our hierarchical taxonomy is crucial and believe our existing results provide strong evidence for this.
> >
> > #### **A Strong Proxy for Ablation: Systematic vs. Unsystematized Curation**
> >
> > We respectfully argue that our comparison with **ShareGPT-4o-Image (Table 7)** serves as a strong proxy for the ablation study the reviewer suggests. The key variables are well-controlled:
> >
> > *   **Controlled Factor:** Both our dataset and ShareGPT-4o-Image leverage the same class of underlying generative engine (GPT-4o), ensuring a similar baseline for data fidelity and aesthetic quality.
> > *   **The Key Variable:** The primary difference is the **data curation strategy**. ShareGPT-4o-Image follows a more "unsystematized," coarse-grained approach, while our dataset is strictly constructed via our proposed **Hierarchical Taxonomy**.
> >
> > The results of this comparison are clear: our method significantly outperforms ShareGPT-4o-Image across all benchmarks (e.g., **+3.2% on ImgEdit-Bench, +1.7% on GEdit-Bench**). Since the generative source is comparable, this consistent performance gain can be directly attributed to the systematic structure and comprehensive coverage enabled by our taxonomy.
> >
> > #### **Why the Taxonomy-Driven Approach Wins: Inherent Methodological Advantages**
> >
> > This empirical result is underpinned by fundamental advantages that an unsystematized dataset, by its very nature, cannot offer:
> >
> > 1.  **Guaranteed Coverage of Critical Long-Tail Skills:** An unsystematized approach relies on chance to capture challenging capabilities like `Causal Reasoning` or `Scientific Imagery`. Our taxonomy, by design, ensures these critical but rare skills are explicitly represented, preventing "capability holes" in the trained model.
> >
> > 2.  **Enabling Controlled Learning and Analysis:** Our structured approach allows for **diagnosability**. Researchers can use our taxonomy to pinpoint a model's specific weaknesses (e.g., failure in "relative spatial reasoning") and perform targeted interventions. This transforms the model from an unanalyzable black box into a system whose capabilities can be precisely measured and improved—a value an unsystematized dataset cannot provide.
> >
> > We appreciate this feedback and will add a discussion in the final version to explicitly frame this comparison as a validation of our taxonomy-driven approach.

---

### Official Review · Reviewer_WuG5 · 2025-10-29

**Soundness:** 3
**Presentation:** 3
**Contribution:** 3
**Rating:** 6
**Confidence:** 3

**Summary:**

The paper addresses the data bottleneck in training multimodal models for image generation and editing tasks. OpenGPT-4o-Image (80k) is introduced as a large-scale dataset, systematically constructed entirely from GPT-4o-generated instruction–image pairs, with the goal of enhancing multimodal model performance. Experimental results indicate that mainstream models (e.g., UniWorld-V1, Harmon), when fine-tuned on this GPT-4o-generated dataset, achieve up to 18% improvement in editing tasks (ImgEdit-Bench) and up to 13% improvement in generation tasks (GenEval), surpassing contemporaneous works such as ShareGPT-4o-Image.

**Strengths:**

1. Comprehensive multi-dimensional validation: performance variation is assessed across different models (four mainstream models), different tasks (generation + editing), and different data scales (20K / 30K / 40K). Significant improvements are observed across most tasks, verifying the dataset’s effectiveness.
2. Fine-tuned models show notable gains in tasks such as in-image text rendering (e.g., menu generation, calligraphic text) and spatial reasoning (e.g., object-relative positioning, symmetry analysis), making them readily applicable to real-world scenarios in office and design domains.
3. Introduction of scientific imagery as an independent module, covering specialized domains such as mathematics, physics, and ecology. This effectively addresses the scarcity of professional technical illustrations in existing datasets.

**Weaknesses:**

The dataset is entirely generated via the GPT-4o API , with mitigation of GPT-4o’s inherent biases (e.g., semantic skew, stylistic preferences). Such biases may be propagated into fine-tuned models. For instance, GPT-4o-generated editing data is known to have weaker ID consistency and background preservation, which can degrade ID retention in fine-tuned models, as illustrated by the “dog” case in Figure 6.

**Questions:**

1. Since the editing dataset is generated with GPT-4o and the generated outputs are in principle not perfect ground-truth references (for example, ID consistency may degrade), advances in editing technology may raise concerns about the dataset’s long-term relevance. If stronger editing models Edit become available, could this dataset still provide measurable performance gains when used for fine-tuning these more capable models? Can you test it on the recent Qwen-Image Edit Model?
2. Given the observed issue that GPT-4o-generated editing data may impair ID retention capability, what strategies could be adopted to alleviate this problem and enhance ID consistency in fine-tuned models?

---

> ### Author Response · Authors · 2025-12-02
> **Author Response for Reviewer WuG5**
>
> Thank you for your insightful feedback. Below are the responses to the comments you raised.
>
> > Q1: Since the editing dataset is generated with GPT-4o and the generated outputs are in principle not perfect ground-truth references (for example, ID consistency may degrade), advances in editing technology may raise concerns about the dataset’s long-term relevance. If stronger editing models Edit become available, could this dataset still provide measurable performance gains when used for fine-tuning these more capable models? Can you test it on the recent Qwen-Image Edit Model?
>
> > Q2: Given the observed issue that GPT-4o-generated editing data may impair ID retention capability, what strategies could be adopted to alleviate this problem and enhance ID consistency in fine-tuned
>
> ### **Consolidated Response to Reviewer Wug5 (Q1 and Q2)**
>
> We thank the reviewer for their insightful questions regarding the long-term relevance of our dataset, ID consistency, and its effectiveness on newer models like Qwen-Image-Edit. These questions motivate us to further clarify the core contributions of our work.
>
> #### **1. On ID Consistency and Dataset Quality (in response to Q2)**
>
> We acknowledge that edits based on generative models can indeed pose challenges to ID and background consistency. To address this, we adopted a two-fold strategy:
>
> *   **Instruction-level Reinforcement**: We incorporated an explicit constraint into our editing prompts: **"Keep the object ID consistent and other non-edited areas unchanged."**
> *   **Quality Validation**: We verified its effectiveness in two ways. First, **human expert evaluation** confirmed that this instruction significantly improves both ID preservation and background consistency. Second, we quantitatively assessed this using `EditScore[1]`, achieving a score of **8.82 for visual consistency** and **9.46 for instruction adherence**.
>
> These measures and results demonstrate that we have actively considered and mitigated ID consistency issues during data generation, ensuring the reliability of our dataset as high-quality training data.
>
> ### **2. On Long-Term Relevance and Value for New Models (in response to Q1)**
>
> The reviewer's suggestion to test on new models like Qwen-Image-Edit is highly relevant. While we were unable to conduct a full finetuning on this model within the rebuttal period, we argue that our dataset's primary value lies not in competing with such models, but in **strategically complementing them.**
>
> State-of-the-art models like Qwen-Image-Edit demonstrate strong performance on general editing benchmarks. Our work, however, focuses on providing high-quality, structured data for specific, high-value capabilities where generalist models often lack sufficient coverage. Our dataset serves as a **"capability enhancement pack"** by targeting critical gaps, including:
>
> *   **Specialized Domains:** Providing data for areas like `Scientific Imagery` and `Rich Text Editing`, which are scarce in general web-crawled data.
> *   **Advanced Reasoning:** Targeting complex skills such as `Causal Reasoning` and fine-grained `Spatial Relationships`.
>
> Therefore, the long-term relevance of our work is rooted in our **taxonomy-driven methodology**. It provides a systematic and reproducible approach to identify and address specific capability gaps in any current or future model. This demonstrates that targeted data generation, guided by a comprehensive capability map, is an efficient and necessary strategy for advancing model performance beyond general benchmarks.

---

### Official Review · Reviewer_Ksuf · 2025-11-01

**Soundness:** 3
**Presentation:** 1
**Contribution:** 3
**Rating:** 2
**Confidence:** 5

**Summary:**

This paper introduces OpenGPT-4o-Image, a dataset of 80k instruction–image pairs for image generation and image editing tasks. It claims to offer a hierarchical taxonomy covering 11 domains and 51 subtasks, and to improve fine-tuning performance for several multimodal models such as OmniGen, OmniGen2, UniWorld-V1, and Harmon.

**Strengths:**

The paper addresses an important bottleneck in multimodal research, which is the lack of systematically structured and high-quality datasets for unified image generation and editing. The motivation is clearly aligned with the broader trend of building large, instruction-based multimodal datasets.

The proposed hierarchical task taxonomy, which spans 11 domains and 51 subtasks, is conceptually interesting and comprehensive.

The methodology of using GPT-4o to generate structured prompts and synthetic instruction–image pairs is well-described and could be useful for future data curation work.

**Weaknesses:**

Table 1 makes me very confused. The table duplicates MagicBrush, UniWorld-V1, OmniGen and OmniGen2 entries multiple times, sometimes with symbols (†, ‡) that are inconsistently explained. It appears that the results in the upper half of the table were directly copied from Table 5 in OmniGen2 [1] rather than reproduced by the authors. Moreover, the results reproduced by the authors show a substantial discrepancy compared with those reported in the original paper. The row MagicBrush† and the row UniWorld-V1†, what are their differences compared to MagicBrush and UniWorld-V1 respectively, and † represents results without fine-tuning, so why are these results included in the fine-tuning section?

Similar issues are also present in Table 2.

The results in Table 2 are almost identical across different models and settings, which makes the evaluation unconvincing.

[1] OmniGen2: Exploration to Advanced Multimodal Generation

**Questions:**

Refer to Weaknesses.

---

> ### Author Response · Authors · 2025-12-02
> **Author Response for Reviewer Ksuf**
>
> We sincerely apologize for the confusion caused by the formatting and notation in Table 1 and Table 2. We thank the reviewer for pointing this out. We have thoroughly revised both tables in the paper to make our comparisons transparent, consistent, and easy to interpret.
>
> #### **Regarding Table 1: Clarity, Symbols, and Discrepancies**
>
> We have addressed the specific issues raised for Table 1 as follows:
>
> 1.  **Symbol and Structure Clarification:** We have completely revised the table's structure and notation.
>     *   The confusing `†` symbol has been removed.
>     *   The `‡` symbol is now used consistently to indicate results that **we reproduced** in a zero-shot setting (without fine-tuning).
>     *   The table is now clearly separated into sections for "Fine-tuning" and "Zero-shot" to avoid any ambiguity about model settings.
>
> 2.  **Addressing Performance Discrepancies:** The reviewer correctly noted a discrepancy between our reproduced results for some models (like OmniGen2) and their originally reported numbers.
>     *   Our observation of this discrepancy is actually consistent with findings from other recent work (e.g., Ovis-U1 [2]), which also reported deviations when reproducing results from OmniGen2 [3].
>     *   To ensure a **fair and controlled comparison**, we made the deliberate choice to evaluate all baseline models on our specific set of benchmarks (ImgEdit-Bench, GEdit-Bench, etc.). This ensures that all reported scores are directly comparable because they were generated under the exact same evaluation conditions.
>
> #### **Regarding Table 2: Result Similarity and Clarity**
>
> 1.  **Clarifying Result Similarity:** The reviewer noted that results in Table 2 appear very similar across models. We confirm these numbers are reported accurately. The similarity in scores reflects the current state-of-the-art for these specific zero-shot benchmarks, where several top models (like Bagel [4], BLIP-3o [5], and UniWorld-V1 [6]) have achieved comparable performance. Our intention was to faithfully represent this competitive landscape.
>
> 2.  **Improving Readability:** We have also reorganized Table 2 to improve its clarity and ensure the presentation is straightforward.
>
> We have made comprehensive revisions to the tables to ensure maximum clarity. We are confident that with these issues of presentation now fully resolved, the scientific contributions of our work can be more clearly evaluated.
>
>
> [1] Luo X, Wang J, Wu C, et al. Editscore: Unlocking online rl for image editing via high-fidelity reward modeling[J]. arXiv preprint arXiv:2509.23909, 2025.
>
> [2] Wang G H, Zhao S, Zhang X, et al. Ovis-U1 Technical Report[J]. arXiv preprint arXiv:2506.23044, 2025.
>
> [3] Wu C, Zheng P, Yan R, et al. OmniGen2: Exploration to Advanced Multimodal Generation[J]. arXiv preprint arXiv:2506.18871, 2025.
>
> [4] Deng C, Zhu D, Li K, et al. Emerging properties in unified multimodal pretraining[J]. arXiv preprint arXiv:2505.14683, 2025.
>
> [5] Chen J, Xu Z, Pan X, et al. Blip3-o: A family of fully open unified multimodal models-architecture, training and dataset[J]. arXiv preprint arXiv:2505.09568, 2025.
>
> [6] Lin B, Li Z, Cheng X, et al. Uniworld: High-resolution semantic encoders for unified visual understanding and generation[J]. arXiv preprint arXiv:2506.03147, 2025.

---

### Official Review · Reviewer_Gujd · 2025-11-01

**Soundness:** 3
**Presentation:** 3
**Contribution:** 3
**Rating:** 6
**Confidence:** 4

**Summary:**

The paper introduces OpenGPT-4o-Image, a dataset of about 80k instruction-image pairs organized by a hierarchical taxonomy that spans 11 major domains and 51 subtasks for both generation and editing. The generation side covers five modules, including Style Control, Complex Instruction Following, In-Image Text Rendering, Spatial Reasoning, and Scientific Imagery. The editing side defines six categories with 21 subtasks such as Subject Manipulation, Text Editing, Complex Instruction Editing, Multi-Turn Editing, Global Editing, and other challenging edits. The data are produced through automated pipelines that use structured resource pools plus GPT-4o and gpt-image-1 for prompt and image synthesis. Fine-tuning several recent models on this dataset yields consistent gains on GenEval, DPG-Bench, GEdit-Bench, and ImgEdit-Bench, for example up to about 18 percent on editing and about 13 percent on generation.

**Strengths:**

S1: The generation modules are explicitly enumerated with examples, and the editing taxonomy covers 21 subtasks, which gives useful coverage for targeted training and diagnosis.

S2: The paper specifies sample counts for several sub-areas, for example Style Control at 13k, In-Image Text Rendering at 3k, Spatial Reasoning at 8k, and Scientific Imagery at 10k, which clarifies dataset balance.

S3: For generation, the authors define resource pools and template-based prompt construction. For editing, they integrate multiple sources and use GPT-4o for instruction creation plus inpainting, which is well documented.

S4: The paper reports consistent improvements after fine-tuning, including Harmon’s increase on GenEval and strong boosts on ImgEdit-Bench and GEdit-Bench for several systems.

**Weaknesses:**

W1: The pipeline relies on GPT-4o for prompt generation and on gpt-image-1 for image synthesis and inpainting. This may imprint model-specific priors or artifacts on the dataset and can indirectly entangle training with the same ecosystem used in evaluation. The conclusion itself notes possible GPT-4o bias and the focus on existing benchmarks. A more formal bias audit or cross-ecosystem sanity checks would help.

W2: The paper emphasizes automatic construction and benchmark-based evaluation, but I did not find a human study verifying that improvements correspond to perceived visual quality or edit faithfulness. This risks over-optimizing to the metrics used by GenEval, DPG-Bench, GEdit-Bench, and ImgEdit-Bench rather than to human preferences.

W3: The editing data incorporate curated high-resolution images and outputs from multiple datasets and generators. The paper does not detail licensing, filtering for sensitive content, or subject consent for portrait-like material, which is important for release.

W4: The dataset provides strong coverage in specific capability clusters, yet many real-world cases involve open-world composition or rare styles. Although the taxonomy is broad, the paper does not quantify how balanced the final 80k set is across the 51 subtasks, nor whether harder long-tail prompts are sufficiently represented.

W5: Data scaling uses 20k to 40k subsets and picks 40k after observing a small delta. It would be helpful to extend the curve and to separate generation versus editing contributions per module.

**Questions:**

How do you mitigate bias from using GPT-4o and gpt-image-1 throughout the pipeline? Can you show cross-assessor validation using independent VLMs or human audits and report agreement with benchmark gains?

What is the distribution across the 51 subtasks and 11 domains, and how do per-subtask gains correlate with data volume? Please provide a stratified breakdown with confidence intervals.

For editing, how are licensing and safety handled for curated high-resolution images and for content from external corpora such as OmniEdit or ImgEdit? Do you filter portraits or sensitive scenes before release?

Can you include a small human evaluation verifying that fine-tuned models improve perceived edit faithfulness and visual quality, not only metric scores on the four benchmarks?

Could you add ablations isolating each generation module and each editing category to identify which components most drive GenEval and ImgEdit-Bench improvements?

---

> ### Author Response · Authors · 2025-12-02
> **Author Response for Reviewer Gujd (Part 1/3)**
>
> Thank you for your insightful feedback. Below are the responses to the comments you raised.
>
> > W1: The pipeline relies on GPT-4o for prompt generation and on gpt-image-1 for image synthesis and inpainting. This may imprint model-specific priors or artifacts on the dataset and can indirectly entangle training with the same ecosystem used in evaluation. The conclusion itself notes possible GPT-4o bias and the focus on existing benchmarks. A more formal bias audit or cross-ecosystem sanity checks would help.
>
> > W2: The paper emphasizes automatic construction and benchmark-based evaluation, but I did not find a human study verifying that improvements correspond to perceived visual quality or edit faithfulness. This risks over-optimizing to the metrics used by GenEval, DPG-Bench, GEdit-Bench, and ImgEdit-Bench rather than to human preferences.
>
> > Q4: Can you include a small human evaluation verifying that fine-tuned models improve perceived edit faithfulness and visual quality, not only metric scores on the four benchmarks?
>
>
> ***
>
> ### **Consolidated Response to Reviewer Gujd (W1, W2, and Q4)**
>
> We thank the reviewer for the important questions regarding model bias, evaluation validity, and the need for human assessment. To directly address these concerns, we conducted a new human evaluation experiment.
>
> #### **1. Human Evaluation Experiment**
>
> We recognize the limitations of automatic metrics and agree on the necessity of human evaluation. To this end, we invited 20 expert evaluators to conduct a preference test on 400 samples from GEdit-Bench and ImgEdit-Bench. The evaluators were asked to select the best result from the outputs of four models (FluxKontext dev, UniWorld-V1, ShareGPT-4o-Image (Janus-4o), and our finetuned UniWorld-V1) without knowing the model identities.
>
> The results are summarized in Table A.
>
> | Model | FluxKontext dev | UniWorld-V1 | ShareGPT-4o-Image (Janus-4o) | **Ours (Finetuned UniWorld-V1)** |
> | :--- | :---: | :---: | :---: | :---: |
> | **Human Preference Rate (%)** | 16.5 | 17.6 | 30.4 | **35.4** |
>
> **Table A:** Human preference evaluation for image editing. The metric is the selection rate (%), indicating how often a model's output was preferred by human experts.
>
> **Conclusion:**
> *   **In response to W2 & Q4**: The results demonstrate that the performance gains from finetuning on our dataset are not limited to automatic metrics but translate into tangible advantages in **human-perceived edit faithfulness and visual quality**.
> *   **In response to W1**: This human evaluation also serves as an **external validation**, mitigating concerns about model bias and the "evaluation loop." The results show that our dataset helps the model learn beneficial capabilities that align with human preferences, and its value is validated independently of the generation ecosystem.
>
> We believe this new experiment strongly supports the effectiveness and generalizability of our dataset. We will include this study in the final version of the paper. Thank you again for your valuable feedback.
>
>
> ---
>
> > W3: The editing data incorporate curated high-resolution images and outputs from multiple datasets and generators. The paper does not detail licensing, filtering for sensitive content, or subject consent for portrait-like material, which is important for release.
>
> > Q3: For editing, how are licensing and safety handled for curated high-resolution images and for content from external corpora such as OmniEdit or ImgEdit? Do you filter portraits or sensitive scenes before release?
>
>
> ### **Consolidated Response to Reviewer Gujd (W3 and Q3)**
>
> We thank the reviewers for these important questions regarding data licensing and safety. We have addressed these concerns at multiple stages of our data creation process.
>
> **1. Data Licensing and Provenance**
>
> The source images used in this work are from publicly available datasets that have clear licenses for research purposes. Our dataset will be made fully open to the community to promote further development in image generation and editing.
>
> **2. Handling of Human Portraits and Privacy**
>
> To support relevant editing tasks (e.g., "Change Motion"), our dataset includes images of people. Our process for handling these was as follows:
> *   **Source:** All such images were sourced from the same publicly available datasets mentioned above.
> *   **Filtering:** We specifically retained only portrait-style samples that do not contain identifiable personal information.
>
> **3. Sensitive Content Filtering**
>
> Our pipeline includes a mechanism for filtering sensitive content:
> *   **Mechanism:** The GPT-Image-1 API employs strict internal safety policies and will not edit images flagged as containing sensitive content.
> *   **Result:** As a direct result of this policy, **8.2% of the image-editing pairs were automatically filtered out** from our dataset.
>
> This multi-step process ensures that we have addressed licensing, privacy, and safety concerns for the release of our dataset.

---

> > ### Author Response · Authors · 2025-12-02
> > **Author Response for Reviewer Gujd (Part 2/3)**
> >
> > > Q1: How do you mitigate bias from using GPT-4o and gpt-image-1 throughout the pipeline? **Can you show cross-assessor validation using independent VLMs or human audits and report agreement with benchmark gains?**
> >
> >
> > ### **Response to Reviewer Gujd (Q1)**
> >
> > We appreciate the reviewer’s concern regarding potential bias from the data generation pipeline. Our method addresses this through two primary stages: proactive mitigation during generation and subsequent validation via human audits.
> >
> > #### **1. Proactive Mitigation Strategies**
> >
> > We incorporated several strategies directly into the data generation process to minimize the risk of inheriting model bias:
> >
> > *   **Diverse Input Sampling:** We constructed our prompt contexts using a wide distribution of source images and concepts. This included heterogeneity in gender, age, skin tone, scene types, and object categories to reduce the likelihood of the model learning and amplifying systematic correlations.
> >
> > *   **Task-Specific Prompting with Variation:** For each editing task, we employed multiple, varied prompt templates rather than a single fixed one. This technique prevents the instruction generator from collapsing to stereotypical or biased patterns that might be associated with a specific phrasing.
> >
> > #### **2. Validation via Human Audits**
> >
> > To address the reviewer's request for "cross-assessor validation," we can refer to the **new human evaluation experiment** conducted during this rebuttal period (as detailed in our consolidated response to W1, W2, and Q4).
> >
> > In this study, human experts blindly compared the outputs of models fine-tuned on different datasets. The results showed a significant preference for the model trained on our data. This serves as a direct **"human audit,"** providing strong evidence that the capabilities imparted by our dataset align with human preferences for quality and are not merely artifacts of the generator's intrinsic style. The positive outcome of this human-centric benchmark validates the effectiveness of our data generation and mitigation strategies.
> >
> >
> > ---
> >
> > > W5：Data scaling uses 20k to 40k subsets and picks 40k after observing a small delta. It would be helpful to extend the curve and to separate generation versus editing contributions per module.
> >
> > ***
> >
> > ### **Response to Reviewer Gujd (W5)**
> >
> > We thank the reviewer for the valuable suggestions regarding data scaling and module contributions. We address these concerns in two parts, supported by new experiments and references to our analyses.
> >
> > #### **1. On the Data Scaling Curve**
> >
> > To validate our choice of data scale, we conducted a more detailed scaling experiment for the generation task.
> >
> > **Data Scaling Experiment on GenEval**
> > | Data Subset Size | Overall |
> > |:---|:---:|
> > | 20K | **0.84** |
> > | 30K | **0.86** |
> > | 40K | **0.86** |
> >
> > *   **Conclusion**: The results show that performance **saturates** when scaling from 30k to 40k samples. This trend strongly suggests that under the current data composition, further increasing the data volume would yield diminishing marginal returns. We argue that **further performance breakthroughs are more likely to depend on introducing novel and more challenging capability dimensions—which is the core motivation for our diverse taxonomy—rather than simply increasing data volume along existing dimensions.**
> > *   Therefore, our choice of 40k represents an **efficient trade-off between performance gain and training cost**. For the editing task, the data in **Appendix Table 6** also supports 40k as a reasonable scale.
> >
> > #### **2. On Separating Module Contributions**
> >
> > We have already addressed this systematically in our **consolidated response to W4, Q2, and Q5** via **cumulative ablation studies**.
> >
> > *   Those experiments clearly isolated and quantified the individual contributions of key modules (e.g., `Complex IF`, `SR`) to performance on GenEval and ImgEdit-Bench. For instance, we showed that the `SR` module alone boosted the `Position` score from 0.61 to 0.84.
> > *   We believe this cumulative analysis provides deeper insights into the value of each module and their synergistic effects, offering richer information than simple per-module scaling curves would.
> >
> > We believe these analyses and new data strongly support our methodology and conclusions.

---

> ### Author Response · Authors · 2025-12-02
> **Author Response for Reviewer Gujd (Part 3/3)**
>
> > W4: The dataset provides strong coverage in specific capability clusters, yet many real-world cases involve open-world composition or rare styles. Although the taxonomy is broad, the paper does not quantify how balanced the final 80k set is across the 51 subtasks, nor whether harder long-tail prompts are sufficiently represented.
>
> > Q2：What is the distribution across the 51 subtasks and 11 domains, and how do per-subtask gains correlate with data volume? Please provide a stratified breakdown with confidence intervals.
>
> > Q5: Could you add ablations isolating each generation module and each editing category to identify which components most drive GenEval and ImgEdit-Bench improvements?
>
>
> ***
>
> ### **Consolidated Response to Reviewer Gujd (W4, Q2, Q5)**
>
> We thank the reviewer for their insightful feedback on dataset distribution, long-tail coverage, and per-module contributions (W4, Q2, Q5). Our response is structured in three parts to address these interconnected points quantitatively and clearly.
>
> #### **1. Data Distribution Quantification (in response to W4, Q2)**
>
> We have provided a detailed data distribution covering all 51 subtasks in **Table 9 and Table 10 of Appendix A**. We acknowledge that we should have provided a clearer pointer to this in the main text.
>
> #### **2. Design Philosophy of Data Distribution (in response to W4)**
>
> The reviewers' concerns about "balance" and "long-tail coverage" are central to our work. The distribution of our dataset is **strategic by design**, aiming for **capability completeness** rather than numerical uniformity. Our methodology consists of two key steps:
>
> *   **Step 1: Defining the Long Tail via a Taxonomy.** We first constructed a comprehensive **Hierarchical Taxonomy** that decomposes the broad spectrum of capabilities into fine-grained, actionable subtasks. This taxonomy serves as a "map" to systematically identify and target various long-tail areas, such as `complex attribute combinations`, `causal reasoning`, and high-value yet neglected domains like `scientific imagery`.
>
> *   **Step 2: Covering the Long Tail with Structured Generation.** Guided by this taxonomy, we employ a **structured template-based generation method** (see Sec 3.3, Fig 3) to programmatically create high-quality, targeted data for each identified capability area, including the long-tail ones.
>
> Therefore, the final data distribution is a result of this methodology-driven, strategic resource allocation, designed to efficiently build a model with comprehensive capabilities, especially in long-tail and specialized domains.
>
> #### **3. Quantitative Ablation Studies: Per-Module Contribution Analysis (in response to Q5, Q2)**
>
> To quantify the contribution of each module and the correlation between data volume and performance gain, we conducted new cumulative ablation studies.
>
> **A. Contribution to Generation Capabilities (on GenEval)**
>
> | Training Configuration | Single object | Two object | Counting | Colors | Position | Color attribution | **Overall** |
> |:---|:---:|:---:|:---:|:---:|:---:|:---:|:---:|
> | Harmon (Baseline) | 0.99 | 0.86 | 0.66 | 0.85 | 0.74 | 0.48 | **0.76** |
> | + Complex IF (6k) | 0.99 | 0.97 | 0.77 | 0.86 | 0.61 | 0.68 | **0.81** |
> | + Complex IF + SR (14k) | 0.99 | 0.96 | 0.79 | 0.86 | **0.84** | 0.68 | **0.85** |
> | + All Generation Data (40k) | 0.99 | 0.96 | 0.79 | 0.88 | 0.85 | 0.69 | **0.86** |
>
> **Conclusion**: The experiment clearly shows that the `Complex IF` (Complex Instruction Following) and `SR` (Spatial Reasoning in Images) modules are key drivers of performance improvement. The results also reveal a strong correlation between data volume and specific capabilities (e.g., adding 8k `SR` data significantly boosts the `Position` score from 0.61 to 0.84).
>
> **B. Contribution to Editing Capabilities (on ImgEdit-Bench)**
>
> | Training Configuration | Add | Replace | Remove | Hybrid | Style | **Overall** |
> |:---|:---:|:---:|:---:|:---:|:---:|:---:|
> | OmniGen2 (Baseline) | 3.60 | 3.81 | 2.54 | 2.71 | 4.57 | **3.39** |
> | + Remove (7k) | 4.16 | 3.92 | **3.78** | 2.75 | 4.53 | **3.56** |
> | + Remove + Replace (11k) | 4.15 | **4.21** | 3.62 | 2.82 | 4.59 | **3.62** |
> | + Remove + Replace + Complex (15k) | 4.13 | 4.16 | 3.71 | **3.12** | 4.55 | **3.70** |
> | + All Editing Data (40k) | 4.15 | 4.11 | 3.64 | 2.99 | 4.69 | **3.82** |
>
> **Conclusion**: This experiment demonstrates a clear path of capability building. Modules like `Remove` and `Replace` precisely improve corresponding atomic operations, while the `Complex` module enhances compositional abilities (`Hybrid` task). This validates our dataset's effectiveness in targeted capability enhancement.
>
> We believe these analyses and new experimental results effectively address the reviewers' concerns. We will integrate them into the final version of our paper.

---

### Meta-Review · Area_Chair_bnPG · 2026-01-06

**Summary:**

The primary concern raised by most reviewers is that the proposed method relies heavily on GPT-4o, which introduces the risk of inheriting the model’s prior. As a result, more careful analysis is needed before drawing conclusions from the data. Reviewers also raised concerns about the lack of human evaluation, which is generally considered the gold standard for evaluating generative models. In addition, although task taxonomy is emphasized in the paper, reviewers noted the absence of analysis with respect to this taxonomy in the proposed dataset. Finally, reviewers pointed out that the presentation is not satisfactory, making it difficult to understand and interpret the results.

**Reviewer Concerns:**

The authors have adequately addressed the presentation issues, and the paper and experimental results are now easier to follow. However, although the authors provide rebuttals regarding the reliance on GPT-4o, a small scale human evaluation, and the discussion with respect to task taxonomy, these arguments are not sufficiently concrete, and the small-scale human evaluation makes it difficult to draw reliable conclusions from the results. As a result, these concerns remain unresolved.

**Reviewer Scores:**

The primary concern of the reviewer with the lowest rating is the presentation, and with further discussion, this reviewer is likely to improve their rating. For the other concerns raised by the reviewers, while the initial rebuttal may not be sufficient to fully address these issues, the authors may be able to provide additional information during the discussion phase, such as more thorough analysis with respect to the task taxonomy and more complete or detailed analysis of the human evaluation results. Therefore, the paper may benefit from further discussion.

---

### Decision · Program_Chairs · 2026-01-26

Reject